artificial intelligence/e-science/statistics

virality, Twitter, sentiment analysis, generalized Waring regression

**Author for correspondence:**
Salud María Jiménez-Zafra
e-mail: sjzafra@ujaen.es

# How do sentiments affect virality on Twitter?

Salud María Jiménez-Zafra[1], Antonio José Sáez-Castillo[2], Antonio Conde-Sánchez[2] and María Teresa Martín-Valdivia[1]

[1]SINAI Department of Computer Science, CEATIC, and [2]Department of Statistics and Operational Research, Universidad de Jaén, Campus Las Lagunillas s/n, Jaén 23071, Spain

SMJ-Z, 0000-0003-3274-8825; AJS-C, 0000-0001-9795-9610;
AC-S, 0000-0003-0450-1585; MTM-V, 0000-0002-2874-0401

Virality on Twitter is catching the attention of researchers, trying to identify factors which increase or decrease the probability of retweeting. We study how terms expressing sentiments affect retweeting frequencies by means of a regression model on the number of retweets, which is specially accurate to deal with virality. We focus on the Spanish political situation during the pseudo-referendum held in Catalonia on 1 October 2017. We have found that the use of negativity in a tweet increases the probability of retweeting and that iSOL lexicon is the one that better determines the relationship between polarity and virality.

## 1. Introduction

Twitter is a micro-blogging service launched in October 2006 that allows users to post 'tweets', which are short messages currently limited to 280 characters. These messages usually include user's opinions and feelings, such as opinions about products and services or political or religious views. Thus, researchers, politicians and the public in general have realized that Twitter is a valuable source of information concerning people's opinions and sentiments. Twitter is becoming one of the main social media in our current society and sometimes it is considered as a thermometer of social problems and events. It has clearly changed how we interact and communicate with each other.

The word of mouth from years ago has been transformed into the dissemination of messages using the 'retweeting' feature of Twitter, which allows users post 'tweets' published by other users. That dissemination is related to two different concepts: first, diffusion, which refers to the number of people who see the message (not available for the researcher) and is driven by factors external to the text (number of author followers, number and popularity of hashtags, etc.); and second, virality, that would measure the effect of the proper message text in the

dissemination of the message, once the effect of the other confounding external variables is eliminated. Both, diffusion and virality are observed by the same outcome, the number of retweets.

So the way of writing something influences its virality. The language used to express an idea, an opinion or simply a fact is fundamental since we are not seeing the other person. Thus, the idea of processing comments or reviews automatically has attracted researchers in the field of text mining (TM) and natural language processing (NLP). Specifically, the area known as sentiment analysis (SA) [1,2] has aroused great interest because it focuses on the automatic identification and analysis of the opinions and emotions expressed in the texts. One of the main tasks in SA is polarity detection, which aims to determine the opinion category that can be assigned to a text. The category can be binary (positive, negative) or it can have different levels of intensity. To address this task, one of the main elements are resources, such as lexicons, which are lists of opinion bearing words that allow the identification of positive and negative words in the texts.

Given the importance of virality and the potential of lexical resources to identify positive and negative sentiments, the main objective of this work is to verify if the use of lexicons can help to assess the virality of a tweet. We focus concretely in a political domain because Twitter is one of the main channels for the dissemination of political messages [3–10]. Specifically, we centre on the context of the Catalan referendum, a very controversial topic in Spain because it was an independence referendum called by the Catalan regional government and suspended by the Constitutional Court of Spain after a request from the Spanish government. For this, we collected a corpus of tweets written in Spanish during the week of the 1 October 2017 using the hashtags #CatalanReferendum and #ReferendumCatalan. Most Spaniards do not agree with Catalonia's independence. Therefore, we have got specific experimental conditions where there is expected to be a greater dissemination of tweets with negative content and a lesser one of positive tweets regardless of other features which contribute to their diffusion. Taking into account this, we have conducted a study to check whether, once we control the diffusion conditions, the use of sentiments terms in a tweet, quantified by means of lexicons, truly affects the probability of retweeting. More specifically, we formulate two hypotheses:

(i) In a political context, the use of opinion bearing words have a significant influence on retweets.
(ii) In the case of the Catalan referendum, the more negative words you use in a tweet, the higher is the probability of retweet; on the contrary, the more positive words the tweet contains, the lower is that probability of retweet.

In order to demonstrate the hypotheses, we statistically assess the effect of some relevant tweet features and the use of sentiment words on the spreading of a tweet. For that, we use a novel regression model which is specially adequate to deal with the confusion between diffusion conditions and virality.

Of course, it is necessary to transform sentiments into measurable variables. In our case, we quantify the effect of negativity and positivity by means of three Spanish sentiment lexicons, NRC, ML-Senticon and iSOL, checking that the latest is the most relevant and predictable resource.

In relation to these lexicons:

(i) NRC Word-Emotion Association Lexicon (EmoLex) [11], one of the sentiment lexicons most currently used that is available in many languages, so it is interesting to see how it behaves.
(ii) ML-Senticon [12], a lexicon for English, Spanish, Catalan, Basque and Galician that is based on SentiWordNet [13,14], which is one of the most referenced resources by the scientific community.
(iii) iSOL [15], a Spanish sentiment lexicon developed by members of our SINAI research group that is an adaptation of Bing Liu's lexicon [16].

From our knowledge, it is the first application wherein iSOL demonstrates its strength as an instrument to identify and quantify the effect of sentiments on the spreading of tweets.

The rest of the paper is organized as follows: we review other studies related to SA, Twitter and virality in Background section; in Methods section, we present the data of study and the model used to assess the effect of sentiment words on Twitter virality; results and discussions are reported in Results section and Discussion section, respectively; finally, Conclusion section summarizes the paper.

# 2. Background

## 2.1. Sentiment analysis on Twitter

Since 2009 the SA research community has started to face the problem of the computational treatment of opinions, sentiments and subjectivity in the short texts of Twitter. Researchers realized that Twitter is a

valuable source of information concerning people's opinions and sentiments from which it was easy to download information to generate corpus and to extract knowledge. Perhaps, one of the first works related to SA and Twitter is the paper of Go *et al.* [17], who analyse the most suitable lexical features to represent a tweet and use different machine-learning approaches to build a classifier for determining the polarity of tweets. They follow the procedure described in [18] in order to build the corpus using emoticons to tag positive and negative posts. After this study, a wide range of methods for SA on Twitter have been published, describing systems with different features and methodologies including machine learning systems (traditional models [19,20] and deep learning models [21,22]), lexicon-based approaches [23–25] and hybrid methods [26,27] (see [28] for a detailed comparison). Moreover, the growing interest of the research community has been reflected in the organization of several workshops which have created benchmark datasets and have enabled direct comparison between different approaches, both as part of the competition and beyond. The most relevant are the International Workshop on Semantic Evaluation[1] (SemEval), whose first edition was held in 2013[2] [29], and the Workshop on Semantic Analysis at SEPLN[3] (TASS) that took place for the first time in 2012 [30].

As it has been mentioned before, lexicon-based methods are one of the approaches used for SA on Twitter. These methods consist of computing the semantic orientation of the words in the text taking into account the positive or negative orientation of words [31]. For this, lexicons are essential, which are lists of opinion bearing words that allow the identification of positive and negative words in texts. In this work, lexicons are also one of the main elements because we study if the use of them can help to predict the virality of a tweet. Specifically, we focus on Spanish tweets, so below we describe the main lexicons used for SA in Spanish: iSOL[4] [15], ML-SentiCon[5] [12] and NRC Word-Emotion Association Lexicon (EmoLex)[6] [11].

The iSOL resource was generated from the Bing Liu English Lexicon [16] by automatically translating it into Spanish and obtaining the Spanish Opinion Lexicon (SOL) resource. Then this resource was manually reviewed in order to improve the final list of words obtaining the improved SOL (iSOL) [15]. The iSOL is composed of 2509 positive and 5626 negative words, thus the Spanish lexicon has 8135 opinion words in total. This resource has been successfully evaluated in several corpora and the results showed that the use of an improved list of sentiment words could be considered a good strategy for unsupervised polarity classification [32–34].

On the other hand, ML-SentiCon is a set of lemma-level sentiment lexicons for English, Spanish and the other three official languages in Spain (Catalan, Basque and Galician). The lexicons were induced using an automatic, semi-supervised method and are formed by eight layers, allowing applications to choose different compromises between the amount of available words and the accuracy of the estimations of their prior polarities. For each PoS-tagged lemma in the resource, two scores are provided: a real value representing the prior polarity, between −1 and 1, and the standard deviation reflecting the ambiguity of that value. According to manual verification of a significant sample, the lexicons for English and Spanish have both high accuracies, over 90% for layers 1–6 and 1–5, respectively [12]. Taking into account the accuracy of the layers, we have considered layers 1–5 that have a total of 702 positive words and 954 negative words.

The NRC Word-Emotion Association Lexicon (EmoLex) [11] is a list of English words associated with one or more emotions (anger, fear, anticipation, trust, surprise, sadness, joy) and to one or more sentiments (positive, negative). This lexicon is also available for more than 100 languages (including Spanish). All these versions have been generated by translating the English terms using Google Translate. In this study, we have used the Spanish part that consists of 2312 positive words and 3324 negative words.

## 2.2. Virality and sentiment analysis

Some previous works have analysed which factors are related to the tweet spreading, considering user features (as the number of followers or the number of friends), specific tweet features (number of

[1]https://semeval.github.io/.

[2]https://www.cs.york.ac.uk/semeval-2013/.

[3]http://tass.sepln.org/.

[4]https://sinai.ujaen.es/index.php/research/resources/isol.

[5]http://www.lsi.us.es/fermin/ML-SentiCon.zip.

[6]https://saifmohammad.com/WebPages/NRC-Emotion-Lexicon.htm.

URLs, hashtags, etc.), aspects about the tweet topic or other characteristics [35–40], although none of these have included sentiments in the tweet between those possible factors.

Other authors have tried to determine if tweets with sentiments or emotions have a greater spreading. For example, Ferrara & Yang [41] study whether sentiments are correlated with the speed and the scope of that spreading, although the sample includes general tweets without URLs or multimedia files, and concludes that tweets with negative emotions are faster spread, whereas tweets with positive emotions have more retweets; nevertheless, no other factors possibly affecting the diffusion process are taken into account.

Bhattacharya *et al.* [42] look for factors related to the number of retweets in the context of a sample from 25 Federal Health Agencies (considering also the time until the first retweet and until the last one), including some of the previously mentioned features and the presence of positive or negative sentiments. Tsugawa & Ohsaki [43] investigate the relation between the sentiment of a message on social media and its virality, defined as the volume and speed of message diffusion, taking into account other factors, such as the number of followers or the existence of URLs or hashtags in the tweet. Similarly, Stieglitz & Dang-Xuan [44] specifically examine whether the affective dimensions of Twitter messages (positive and negative sentiments) associated with political parties or politicians are related to retweet behaviour in terms of (i) quantity of triggered retweets and (ii) speed of retweeting defined by the time lag between the original tweet and the first retweet, considering similar characteristics to Tsugawa & Ohsaki [43]. In the context of a terrorism attack, Burnap *et al.* [45] establish several hypothesis about the factors involved in the virality of a tweet, in terms of both the volume and the survival time; between them, they claim that, fixing the topic, tweets with negative sentiments will be positively related with the number of retweets.

In this paper, we try to demonstrate that sentiment words significantly affect the spreading of a tweet. More specifically, we will check that the use of negativity contributes to the retweeting average in the political domain, while positivity reduces the probability of retweeting.

# 3. Methods

In this section, we present the procedure followed to collect the data of study, the features used to represent each tweet and the statistical model employed to assess the effect of the use of sentiment words on the virality of a tweet.

## 3.1. Data collection

We employed the Twitter Standard Search API v. 1.1[7] to identify all the tweets released on 1 October 2017 with the hashtags `#CatalanReferendum` or `#ReferendumCatalan`. Later, we collected features of these tweets on 31 October 2017 in order to analyse their virality. Using this method, a total of 46 962 tweets from 25 847 Twitter accounts were collected. This dataset [46] is available at https://doi.org/10.5061/dryad.stqjq2c24.

Each tweet of this collection consists of attributes provided by the Twitter API, which will be explained in §3.2, plus the following 10 attributes that we have added:

— *hashtags*: Total number of hashtags in the tweet.
— *urls*: Total number of URLs in the tweet.
— *mentions*: Total number of users mentioned in the tweet.
— *time*: Interval of the day on which the tweet was published (morning (06.00–12.00), afternoon (12.00–18.00), evening (18.00–00.00) or night (00.00–06.00)).
— *pos_iSOL*: Total number of positive words found in the tweet using iSOL lexicon.
— *neg_iSOL*: Total number of negative words found in the tweet using iSOL lexicon.
— *pos_NRC*: Total number of positive words found in the tweet using NRC lexicon.
— *neg_NRC*: Total number of negative words found in the tweet using NRC lexicon.
— *pos_mlS*: Total number of positive words found in the tweet using ML-SentiCon lexicon.
— *neg_mlS*: Total number of negative words found in the tweet using ML-SentiCon lexicon.

In order to compute the total number of positive and negative words using the lexicons iSOL, NRC and ML-SentiCon, tweets were preprocessed in the following way: they were tokenized using NLTK

---

[7]https://developer.twitter.com/en/docs/tweets/search/api-reference/get-search-tweets.html.

**Table 1.** Examined features.

| tweet-specific | |
| --- | --- |
| retweet | # of retweets recorded for a given tweet |
| favourite | # of favourites recorded for a given tweet |
| quote | whether a tweet includes a quote of other tweet |
| reply | # whether a tweet is in reply to other tweet |
| hashtags | # of hashtags in a tweet |
| urls | # of URLs in a tweet |
| mentions | # of usernames specified in a tweet |
| time | time interval when the tweet was created (categorical variable)[a] |
| pos_iSOL | # of positive words (iSOL) |
| neg_iSOL | # of negative words (iSOL) |
| pos_NRC | # of positive words (NRC) |
| neg_NRC | # of negative words (NRC) |
| pos_mlS | # of positive words (ML-SentiCon) |
| neg_mlS | # of negative words (ML-SentiCon) |
| **user-level** | |
| verified | whether the tweet is from a verified user |
| followers | # of users who follow the author of a tweet |
| friends | # of friends that the author is following |
| listed | # of lists which include the author of a tweet |
| user_fav | # of favourited tweets by a user |
| statuses | # of tweets made by the author since the creation of the account |
| months | # of months since the creation of the account |

[a]Morning (6.00–12.00), afternoon (12.00–18.00), evening (18.00–0.00) or night (0.00–6.00)

TweetTokenizer;[8] all letters were converted to lower-case and those accented were replaced by the corresponding unaccented letter.

## 3.2. Tweet features

Table 1 lists the features we have included to represent each tweet, under two categories: user-level (e.g. numbers of followers and friends) and tweet-specific features, such as the number of retweets, positive or negative terms in the tweet, etc.

As we mentioned, 46 962 tweets were collected, 1.91% of them from verified users; 8.08% were tweeted in the morning (from 6.00 to 12.00), 55.24% in the afternoon (from 12.00 to 18.00), and 36.27% in the evening (from 18.00 to 0.00); 12.62% included a quote of another tweet and 6.85% of them were in reply to another tweet. Table 2 shows descriptive statistics of the rest of features.

The number of retweets oscillates between 0 and 26 212, with a mean of 8.92, clearly below the variance, and a strong right asymmetry with a heavy tail; these results are similar to those published in previous studies [43,45]. Furthermore, 63.31% of tweets are not retweeted, only 5.03% are retweeted more than 10 times and a wide spreading of more than 100 retweets only affects 1.04% of tweets, also agreeing other previous studies [42].

In reference to the number of positive and negative words in tweets, it is remarkable that a very high percentage of them do not present any sentiment. On the one hand, iSOL do not find any positive word in 76.24% of tweets, being 66.76% and 93.72% this percentage with NRC and ML-SentiCon, respectively. On the other hand, the percentage of tweets with no negative words detected is 59.92% with iSOL, 67.74% with NRC and 87.67% with ML-SentiCon. Table 2 shows that the maximum number of positive terms

[8]http://www.nltk.org/api/nltk.tokenize.html.

**Table 2.** Descriptive statistics of numerical features.

|  | min | max | mean | median | s.d. |
|---|---|---|---|---|---|
| retweet | 0 | 26 212 | 8.92 | 0 | 214.31 |
| favourite | 0 | 34 324 | 9.06 | 0 | 244.61 |
| hashtags | 1 | 10 | 1.79 | 1 | 1.13 |
| urls | 0 | 3.00 | 0 | 0 | 0.46 |
| mentions | 0 | 8 | 0.28 | 0 | 0.66 |
| pos_iSOL | 0 | 5 | 0.28 | 0 | 0.56 |
| neg_iSOL | 0 | 8 | 0.55 | 0 | 0.79 |
| pos_NRC | 0 | 6 | 0.43 | 0 | 0.69 |
| neg_NRC | 0 | 6 | 0.42 | 0 | 0.70 |
| pos_mlS | 0 | 4 | 0.07 | 0 | 0.26 |
| neg_mlS | 0 | 5 | 0.14 | 0 | 0.41 |
| followers | 0 | 6 408 669 | 11536.50 | 298 | 175546.78 |
| friends | 0 | 483 424 | 1178.87 | 383 | 5989.94 |
| listed | 0 | 54 377 | 113.54 | 5 | 1400.75 |
| user_fav | 0 | 770 740 | 6220.21 | 1067 | 23278.15 |
| statuses | 1 | 7 930 051 | 22827.33 | 4692 | 85552.64 |
| months | 1 | 134 | 59.86 | 68 | 30.39 |

**Table 3.** Kendall's tau coefficient for the number of positive words.

|  | pos_iSOL | pos_NRC | pos_mlS |
|---|---|---|---|
| pos_iSOL | 1.00000 | 0.24798 | 0.37931 |
| pos_NRC |  | 1.00000 | 0.13987 |
| pos_mlS |  |  | 1.00000 |

in a tweet is 5 for iSOL, 6 for NRC and 4 in the case of ML-SentiCon, whereas the maximum number of negative words is 8, 6 and 5, respectively. Finally, tables 3 and 4 include Kendall's tau coefficient for positive and negative words in the three lexicons, showing weak relations between them.

## 3.3. Model

We perform a regression analysis to demonstrate the relation between message sentiment, identified by the three lexicons, and retweeting once the effects of the rest of the factors affecting the message diffusion are eliminated.

Specifically, we fit three regression models wherein the dependent variable is `retweet` and the covariates were the rest of features shown in table 1, using one of the three lexicons to identify positive and negative sentiments in each case. The variables `favourite`, `followers`, `friends`, `listed`, `user_fav` and `statuses` present a wide range, which suggest to employ a log-scale. Dichotomous covariates, that is, `quote`, `reply` and `verified` are considered codifying FALSE as zero and TRUE as one, and `time` is converted into three dummy dichotomous covariates considering the morning as the reference level.

Since the dependent variable is a count variable, we have considered adequate models for it. In this sense, the number of retweets is extremely overdispersed, i.e. the variance is much greater than the mean, so a typical Poisson (P) regression model is completely inadequate; on the other hand, regardless that previous works [35,37–40,43,44] employ a negative binomial (NB) regression model, we have considered a generalized Waring (GW) regression model, which provides a better fit to data: in

**Table 4.** Kendall's tau coefficient for the number of negative words.

|  | neg_iSOL | neg_NRC | neg_mlS |
|---|---|---|---|
| neg_iSOL | 1.00000 | 0.45122 | 0.41194 |
| neg_NRC |  | 1.00000 | 0.27833 |
| neg_mlS |  |  | 1.00000 |

general, this distribution is more adequate than the negative binomial when data are affected by a strong overdispersion [47,48], which is the case of retweeting.

These count data models commonly establish a log-linear relation between the average of the dependent variable, $Y$, and the covariates, $X_1, \ldots, X_k$, from

$$\mu_{Y|X_1=x_1,\ldots,X_k=x_k} = \exp\left(\beta_0 + \beta_1 \times x_1 + \cdots + \beta_k \times x_k\right), \tag{3.1}$$

where $\mu_{Y|X_1=x_1,\ldots,X_k=x_k}$ denotes the conditional average of the dependent variable given the current values of the covariates. This equation permits to interpret regression coefficients $\beta_j$ in relation to the covariates effect depending on whether they are introduced in the model linearly or in log-scale:

(i) From equation (3.1), the relative increase of the average of the dependent variable when a covariate introduced in the model linearly is increased in one unit and the rest remains constant is given by

$$\frac{\mu_{Y|X_1=x_1,\ldots,X_j=x_j+1,\ldots,X_k=x_k}}{\mu_{Y|X_1=x_1,\ldots,X_j=x_j,\ldots,X_k=x_k}} = \exp\left(\beta_j\right). \tag{3.2}$$

(ii) On the other hand, if $X_j$ is one of the features which have been introduced in the model in log-scale, a relative increase of 1% in $X_j$ implies an approximate increase of the average of the dependent variable $\beta_j$% [49].

Thus, in our case, regression coefficients $\beta_i$ permit a quantification of the effect of a variation of each feature on the average of the number of retweets, with the rest of features remaining constant, that is, regardless of their influence. In particular, $\beta_i$ coefficients corresponding to the number of positive or negative words show the effect of sentiment of a tweet in the number of retweets, once the effect of the rest of features related to diffusion conditions is controlled, that is, represent the effect of polarity on virality.

An alternative methodology [42] separates data in two parts: on the one hand, a logistic binary model tries to predict the probability of a tweet to be retweeted; on the other hand, a truncated at zero model describes tweets with at least one retweet. We have not found empirical or theoretical evidence for that mixing treatment.

We have compared count data regression models using a standard goodness of fit measure such as the Akaike information criterion (AIC) [50]. In all the fitted models, the GW had clearly the lowest AIC value compared with the P and the NB models, showing a better fit.

Calculations have been carried out with GWRM [51] and MASS [52] libraries of R software [53].

## 4. Results

Table 5 includes regression coefficient estimates and significance of the features in the three GW fitted models, which include iSOL, NRC and ML-SentiCon describing sentiments, respectively. AIC value of the model fitted with iSOL (107644.8) is slightly lower in comparison with the models fitted with NRC and ML-SentiCon (107675.1 and 107647.7, respectively), indicating a better fit to data. Analogous NB models show AIC values of 108553.4, 108594.2 and 108569.9, respectively, indicating a poorer adequacy to data.

The most remarkable result from table 5 is that, whereas models employing NRC and ML-SentiCon denote that both the number of positive and negative words have a significant negative effect on the average of the number of retweets, the model fitted with iSOL indicates that each new positive word implies a relative change of $\exp(-0.074) = 0.9286$, that is, a decrease of 7.14%, in the average of the number of retweets; and each new negative word implies a relative change of $\exp(0.034) = 1.0346$, that is, an increase of 3.46%, in the average of the number of retweets, always comparing with tweets with similar other features.

**Table 5.** Regression coefficient estimates in the fitted GW models with each lexicon.

| features | $\hat{\beta}_i$ and statistical significance | | |
| --- | --- | --- | --- |
| | iSOL | NRC | ML-SentiCon |
| (intercept) | −2.363*** | −2.330*** | −2.336*** |
| positive_words | −0.074*** | −0.040*** | −0.104*** |
| negative_words | 0.034*** | −0.023* | −0.109*** |
| log(favourite + 1) | 1.263*** | 1.261*** | 1.261*** |
| reply | −0.326*** | −0.329*** | −0.328*** |
| quote | −0.051* | −0.057* | −0.050 |
| hashtags | 0.080*** | 0.077*** | 0.078*** |
| urls | 0.329*** | 0.325*** | 0.323*** |
| mentions | 0.037** | 0.035** | 0.036** |
| time_morning | 0.058* | 0.058* | 0.055* |
| time_afternoon | −0.054 | −0.058* | −0.061* |
| time_evening | −0.064 | −0.082 | −0.090 |
| verified | −0.268*** | −0.260*** | −0.261*** |
| log(followers+1) | 0.113*** | 0.113*** | 0.112*** |
| log(friends + 1) | 0.040*** | 0.039*** | 0.039*** |
| log(listed + 1) | 0.045*** | 0.045*** | 0.044*** |
| log(user_fav + 1) | −0.032*** | −0.032*** | −0.031*** |
| log(statuses) | 0.012 | 0.013* | 0.013* |
| months | −0.002*** | −0.002*** | −0.002*** |

\*\*\*Significant at 0.1%; \*\*Significant at 1%; \*Significant at 5%.

In relation to the rest of features, the model fitted using iSOL establishes:

— `favourite`, `hashtags`, `urls`, `mentions`, `followers`, `friends` and `listed` have significant positive effects on the average number of retweets. Especially, it is remarkable that an increase of 1% in `favourites + 1`, with the rest of the features remaining constant, will suppose an approximate increase of 1.26% in the average number of retweets; in the same way, if `urls` increases in one and the rest of features remain constant, the relative change in the average of retweets will be exp (0.329) = 1.3896, which supposes an increase of 38.96%.

— `reply`, `verified`, `user_fav` and `months` have a significant negative effect on the number of retweets. Between them, the highest effects correspond to `reply` and `verified`: in the first case, since exp (−0.326) = 0.7218, the model establishes that tweets in reply have 27.82% lower average number of retweets than those not in reply and similar remaining features; in the second case, since exp (−0.268) = 0.7649, the model indicates that tweets from verified accounts are retweeted 23.51% less than those from non-verified accounts and similar other features.

— `statuses` presents a non-significant effect on the average number of retweets.

— There are no significant differences in the average number of retweets between tweets emitted in the night and those emitted in the morning, or in the morning in comparison with the evening. On the contrary, there is a significant difference between tweets sent out in the afternoon and in the morning.

Finally, we find that models fitted with NRC and ML-Senticon provide quite similar results about the effect of these features on the number of retweets.

## 5. Discussion

Taking into account the results, we consider that the integration of iSOL semantic resource can be used as a significant predictor related to the virality for the number of retweets, once the effect of other tweets features related to its diffusion are isolated.

From our point of view, the most relevant result in this study is the finding of an inverse effect of the negative and positive sentiments detected through iSOL lexicon on the average of retweets in the context of the tweets published after the pseudo-referendum in Catalonia. Taking into account that we only collected Spanish tweets, it is possible to think that the majority of the population showed displeasure about the referendum issue. The fact that the model fitted with iSOL lexicon determines that including positive terms contributes to a lower average of retweets and, similarly, the inclusion of negative terms in a tweet corresponds to a higher virality, seems to suppose a confirmation of this hypothesis.

If we compare this finding with previously mentioned results where it has been analysed how sentiments affect the virality of a tweet, we find that this effect depends on the analysed context, although, in general, in none of them there is such a clear differentiation of the effects of positivity and negativity on virality. Thus, Tsugawa & Ohsaki [43], who analyse tweets without a specific topic, determine that if the tweet has positive or negative sentiments, the number of retweets increases, although the effect is greater for those who have negative sentiments. It should be noted that they do not consider tweets that have not been retweeted, so the distribution of $Y$ is truncated, a matter that is not taken into account by the model considered. Similarly, Burnap *et al.* [45] analyse tweets in the context of a terrorist attack. They postulate that tweets with negative sentiments and high levels of tension will be positively related to the number of retweets, but it is determined that tweets with positive content increase the number of retweets. In this case, it has been considered a truncated model in 0, although the response variable is greater than or equal to 6, which may have affected the results obtained. In the work of Stieglitz & Dang-Xuan [44], two datasets taken before two elections in Germany are analysed: in one of them it is obtained that positive tweets have a positive effect on the number of retweets, while in the other tweets with negative sentiments are retweeted more than the positive ones. Meanwhile, Bhattacharya *et al.* [42] consider a hurdle model, so we must distinguish the effect on the logistic part and on the counting part. Thus, on the logistic part, it is obtained that tweets with positive and negative sentiments have a negative effect on retweets (stronger for tweets with negative sentiments). However, for the counting part only tweets with negative sentiments have a significant negative effect.

It should also be noted that only iSOL has succeeded in discriminating the inverse effect of positivity and negativity on the average of retweets, while the models fitted with NRC and MLSenticon do not differentiate the effect of sentiments in the spreading of a tweet. It must be taken into account that iSOL was built from the automatic translation of the Bing Liu's lexicon with subsequent manual supervision to improve it, while NRC and MLSenticon did not have that manual supervision. The differences in the generation of these lexicons is reflected in their classification form. In fact, the measure of correlation using Kendall's tau coefficient (tables 3 and 4) indicates that the relationship between the three lexicons is scarce, especially in the identification of positive terms.

# 6. Conclusion

Lexicons are resources that allow the identification of positivity and negativity in texts. Our results demonstrate that iSOL is able to identify sentiments in a significantly different way than ML-Senticon and NRC Word-Emotion Association Lexicon (EmoLex). The results obtained with iSOL show that negativity increases the virality of a tweet in the context of the Catalan referendum, whereas positivity decreases it. On the contrary, the other two lexicons do not differentiate the effect of both sentiments in the spreading of a tweet. This could be explained because although iSOL was generated from the automatic translation of Bing Liu's lexicon, it was manually reviewed in order to improve its quality. However, ML-Senticon and NRC Word-Emotion Association Lexicon (EmoLex) were not supervised manually. This fact and the results obtained in previous experiments [32–34] confirm that iSOL is a good resource for SA in Spanish.

Data accessibility. Jiménez-Zafra, Salud María; Martín-Valdivia, María Teresa; Sáez-Castillo, Antonio José; Conde-Sánchez, Antonio (2020), Catalan Referendum Twitter corpus, Dryad, Dataset, https://doi.org/10.5061/dryad.stqjq2c24 [46].
Authors' contributions. S.M.J.-Z. collected the dataset, computed sentiment analysis features and drafted the manuscript. A.J.S.-C. designed and performed the regression analysis and drafted the manuscript. A.C.-S. designed and performed the regression analysis and drafted the manuscript. M.T.M.-V. supervised the study and drafted the manuscript. All authors read and approved the final manuscript.
Competing interests. We declare we have no competing interests.

Funding. This work has been partially supported by a grant from Fondo Social Europeo, Administration of the Junta de Andalucía (DOC_01073), Ministerio de Educación Cultura y Deporte (MECD – scholarship FPU014/00983), Fondo Europeo de Desarrollo Regional (FEDER) and LIVING-LANG project (RTI2018-094653-B-C21) from the Spanish Government.

Acknowledgements. A.J.S.-C. thanks you for your friendship and involvement in everything you did. A warm hug to heaven.

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
