## [Peer Review File · Royal Society Open Science]

Review History

RSOS-201756.R0 (Original submission)

Review form: Reviewer 1

Is the manuscript scientifically sound in its present form?

Yes

Are the interpretations and conclusions justified by the results?

Yes

Is the language acceptable?

Yes

Do you have any ethical concerns with this paper?

No

Have you any concerns about statistical analyses in this paper?

No

Recommendation?

Accept with minor revision (please list in comments)

Comments to the Author(s)

This paper deals with virality on Twitter. Specifically, the authors present a study about how sentiment terms affect the probability of retweeting. They analyze what sentiment makes a tweet more likely to be retweeted. For this, they propose a regression model on the number of retweets that examines the most used Spanish lexicons. As a data set to carry out the study they compile a corpus of tweets related to the pseudo referendum held in Catalonia on October 1st, 2017.

The review of the state of the art is appropriate, distinguishing the two relevant parts of the study: sentiment analysis on Twitter and, virality and sentiment analysis. However, I miss lexicon references in the following sentence: "Specifically, we focus on Spanish tweets so below we describe the main lexicons used for SA in Spanish: iSOL, ML-SentiCon and NRC Word-Emotion Association Lexicon (EmoLex)". Please, include references to the description papers and download links. I also miss references to SemEval and TASS workshops. Please, include them. Regarding the different methods for sentiment analysis on Twitter, you must include deep learning.

Regarding the Methods section, it presents the dataset, the features used in the study and the regression model. The description of the dataset is clear and concise, but it is not specified if it is accessible. Please, include download link. On the other hand, in relation to the features used, they are clearly explained, analyzed and summarized in a table, which facilitates replicability. At last, this section shows all the details of the regression model proposed in this study which is especially adequate to deal with the confusion between diffusion conditions and virality. In Table 5, there are some features in Spanish. Please replace with their corresponding English

Finally, authors present results and a discussion thereof. The most interesting aspects are that negative terms increase virality of a tweet in the context of the Catalan referendum, and that iSOL lexicon is a good resource for determining the relationship between sentiment and virality in Spanish texts.

Decision letter (RSOS-201756.R0)

Dear Dr Jiménez-Zafra

On behalf of the Editors, we are pleased to inform you that your Manuscript RSOS-201756 "How Do Sentiments Affect Virality on Twitter?" has been accepted for publication in Royal Society Open Science subject to minor revision in accordance with the referees' reports. Please find the referees' comments along with any feedback from the Editors below my signature.

Please submit your revised manuscript and required files (see below) no later than 7 days from today's (ie 09-Mar-2021) date. Note: the ScholarOne system will 'lock' if submission of the

revision is attempted 7 or more days after the deadline. If you do not think you will be able to meet this deadline please contact the editorial office immediately.

Best regards,

on behalf of Professor Mirella Lapata (Associate Editor) and Marta Kwiatkowska (Subject Editor)
openscience@royalsociety.org

Editor Comments to the Author:

Firstly, please accept our apologies for the unusual delays incurred during peer-review of your manuscript. The pandemic coupled with the festive period unfortunately made it far more difficult than usual to secure referees. We are pleased to accept your manuscript subject to minor revisions, however. Please ensure to supply a thorough point-by-point response to the comments received, as well as a tracked changes version of your revised paper. We look forward to receiving your revision.

Reviewer comments to Author:

Reviewer: 1
Comments to the Author(s)

This paper deals with virality on Twitter. Specifically, the authors present a study about how sentiment terms affect the probability of retweeting. They analyze what sentiment makes a tweet more likely to be retweeted. For this, they propose a regression model on the number of retweets that examines the most used Spanish lexicons. As a data set to carry out the study they compile a corpus of tweets related to the pseudo referendum held in Catalonia on October 1st, 2017.

The review of the state of the art is appropriate, distinguishing the two relevant parts of the study: sentiment analysis on Twitter and, virality and sentiment analysis. However, I miss lexicon references in the following sentence: "Specifically, we focus on Spanish tweets so below we describe the main lexicons used for SA in Spanish: iSOL, ML-SentiCon andNRC Word-Emotion Association Lexicon (EmoLex)". Please, include references to the description papers and download links. I also miss references to SemEval and TASS workshops. Please, include them. Regarding the different methods for sentiment analysis on Twitter, you must include deep learning.

Regarding the Methods section, it presents the dataset, the features used in the study and the regression model. The description of the dataset is clear and concise, but it is not specified if it is accessible. Please, include download link. On the other hand, in relation to the features used, they are clearly explained, analyzed and summarized in a table, which facilitates replicability. At last, this section shows all the details of the regression model proposed in this study which is especially adequate to deal with the confusion between diffusion conditions and virality. In Table 5, there are some features in Spanish. Please replace with their corresponding English

Finally, authors present results and a discussion thereof. The most interesting aspects are that negative terms increase virality of a tweet in the context of the Catalan referendum, and that iSOL lexicon is a good resource for determining the relationship between sentiment and virality in Spanish texts.

===PREPARING YOUR MANUSCRIPT===

===PREPARING YOUR REVISION IN SCHOLARONE===

Author's Response to Decision Letter for (RSOS-201756.R0)

See Appendix A.

Decision letter (RSOS-201756.R1)

Dear Dr Jiménez-Zafra,

I am pleased to inform you that your manuscript entitled "How Do Sentiments Affect Virality on Twitter?" is now accepted for publication in Royal Society Open Science.

Best regards,

on behalf of Professor Mirella Lapata (Associate Editor) and Marta Kwiatkowska (Subject Editor)
openscience@royalsociety.org

Appendix A

February 15, 2021

Re: Royal Society Open Science - Decision on Manuscript ID RSOS-201756

Dear editor,

I send you the reviewed version of the paper identified with the ID RSOS-201756, entitled “How Do Sentiments Affect Virality on Twitter?” and written by Salud María Jiménez-Zafra, Antonio Sáez-Castillo, Antonio Conde-Sánchez and María Teresa Martín-Valdivia.

We would like to thank you for the reviews of our article and for your own assessment. They have helped us frame the paper better and elaborate on aspects that were unclear. Here we explain how we have addressed the reviewers’ comments and how we have improved the paper.

In what follows, the original comments from reviewers are in normal font. Our responses are in blue and indented.

Best regards,

Salud M. Jiménez-Zafra (On behalf of all authors)

1 Editor’ comments

Firstly, please accept our apologies for the unusual delays incurred during peer-review of your manuscript. The pandemic coupled with the festive period unfortunately made it far more difficult than usual to secure referees. We are pleased to accept your manuscript subject to minor revisions, however. Please ensure to supply a thorough point-by-point response to the comments received, as well as a tracked changes version of your revised paper. We look forward to receiving your revision.

We are pleased to receive acceptance with a minor revision. Here we provide a thorough point-by-point response to the comments received. All changes in the document are in red font, to help to see where they have been applied.

2 Reviewer #1

This paper deals with virality on Twitter. Specifically, the authors present a study about how sentiment terms affect the probability of retweeting. They analyze what sentiment makes a tweet more likely to be retweeted. For this, they propose a regression model on the number of retweets that examines the most used Spanish lexicons. As a data set to carry out the study they compile a corpus of tweets related to the pseudo referendum held in Catalonia on October 1st, 2017.

The review of the state of the art is appropriate, distinguishing the two relevant parts of the study: sentiment analysis on Twitter and, virality and sentiment analysis. However, I miss lexicon references in the following sentence: “Specifically, we focus on Spanish tweets so below we describe the main lexicons used for SA in Spanish: iSOL, ML-SentiCon and NRC Word-Emotion Association Lexicon (EmoLex)”. Please, include references to the description papers and download links.

Thank you for pointing this out. We have included the references to the description papers and the download links.

”Specifically, we focus on Spanish tweets so below we describe the main lexicons used for SA in Spanish: iSOL¹ [15], ML-SentiCon²[12] and NRC Word-Emotion Association Lexicon (EmoLex)³[11].”

I also miss references to SemEval and TASS workshops. Please, include them.

References to SemEval and TASS workshops have been included as it is shown below.

¹<https://sinai.ujaen.es/index.php/research/resources/isol>

²<http://www.lsi.us.es/fermin/ML-SentiCon.zip>

³<https://saifmohammad.com/WebPages/NRC-Emotion-Lexicon.htm>

”The most relevant are the International Workshop on Semantic Evaluation⁴ (SemEval), whose first edition was held in 2013⁵ [27], and the Workshop on Semantic Analysis at SEPLN⁶ (TASS) that took place for the first time in 2012 [28].”

Regarding the different methods for sentiment analysis on Twitter, you must include deep learning.

Machine learning systems have been divided into traditional models and deep learning models.

”After this study, a wide range of methods for SA on Twitter have been published, describing systems with different features and methodologies including machine learning systems (traditional models [19,20] and deep learning models [21,22])...”

Regarding the Methods section, it presents the dataset, the features used in the study and the regression model. The description of the dataset is clear and concise, but it is not specified if it is accessible. Please, include download link.

The reference to the dataset and the download link have been included.

”Using this method, a total of 46962 tweets from 25847 Twitter accounts were collected. This dataset [46] is available at <https://doi.org/10.5061/dryad.stjq2c24>.”

[46] Jiménez-Zafra SM, Martín-Valdivia MT, Sáez-Castillo AJ, Conde-Sánchez A. Catalan Referendum Twitter corpus, Dryad, Dataset; 2020. Available from: <https://doi.org/10.5061/dryad.stjq2c24>

On the other hand, in relation to the features used, they are clearly explained, analyzed and summarized in a table, which facilitates replicability. At last, this section shows all the details of the regression model proposed in this study which is especially adequate to deal with the confusion between diffusion conditions and virality. In Table 5, there are some features in Spanish. Please replace with their corresponding English.

Features in Spanish have been replaced by their corresponding in English.

`time_morning` 0.058 * 0.058 * 0.055 *
`time_afternoon` -0.054 -0.058 * -0.061 *
`time_evening` -0.064 -0.082 -0.090

Finally, authors present results and a discussion thereof. The most interesting aspects are that negative terms increase virality of a tweet in the context of the Catalan referendum, and that iSOL lexicon is a good resource for determining the relationship between sentiment and virality in Spanish texts.

Thank you very much for all the comments that have helped to improve our paper.

⁴<https://semeval.github.io/>

⁵<https://www.cs.york.ac.uk/semeval-2013/>

⁶<http://tass.sepln.org/>